# UP-DP: Unsupervised Prompt Learning for Data Pre-Selection with Vision-Language Models

Xin Li    Sima Behpour    Thang Doan    Wenbin He    Liang Gou    Liu Ren

Bosch Research North America, Bosch Center for Artificial Intelligence (BCAI)

{xin.li9, sima.behpour, thang.doan, wenbin.he2, liang.gou, liu.ren}@us.bosch.com

## Abstract

In this study, we investigate the task of data pre-selection, which aims to select instances for labeling from an unlabeled dataset through a single pass, thereby optimizing performance for undefined downstream tasks with a limited annotation budget. Previous approaches to data pre-selection relied solely on visual features extracted from foundation models, such as CLIP and BLIP-2, but largely ignored the powerfulness of text features. In this work, we argue that, with proper design, the joint feature space of both vision and text can yield a better representation for data pre-selection. To this end, we introduce UP-DP, a simple yet effective unsupervised prompt learning approach that adapts vision-language models, like BLIP-2, for data pre-selection. Specifically, with the BLIP-2 parameters frozen, we train text prompts to extract the joint features with improved representation, ensuring a diverse cluster structure that covers the entire dataset. We extensively compare our method with the state-of-the-art using seven benchmark datasets in different settings, achieving up to a performance gain of 20%. Interestingly, the prompts learned from one dataset demonstrate significant generalizability and can be applied directly to enhance the feature extraction of BLIP-2 from other datasets. To the best of our knowledge, UP-DP is the first work to incorporate unsupervised prompt learning in a vision-language model for data pre-selection.

## 1   Introduction

Data-efficient machine learning, which aims to identify the best subsets of data samples (to either label or not) in order to achieve optimal model performance, has been a crucial research field in the era of data-hungry deep learning [12, 31, 24, 40, 9]. In this work, we focus on a new and practical task of **data pre-selection** for data-efficient visual object recognition (Fig.1-a). The goal of data pre-selection is to select instances for labeling from an unlabeled dataset through *a single pass* to maximize model performance for **unknown** downstream vision tasks (e.g., no knowledge about prediction categories, shown in Fig.1-b), given a limited annotation budget. This task is motivated by two practical needs: at the data acquisition stage, we want to collect minimal data to support potentially diverse downstream tasks (e.g., classification, action recognition, or even detection for videos or images in Fig.1-b) and training paradigms (e.g. active learning, semi-supervised learning, supervised learning in Fig.1-c); at the annotation stage, we aim to achieve good performance with fewer human annotations over the pre-selected data.

Traditionally, to tackle the challenge of data efficiency learning, there are two main approaches: Semi-supervised learning (SSL) [3, 35, 21] and Active Learning (AL) [34, 10, 2]. SSL aims to address the problem of label scarcity by leveraging a limited quantity of labeled data in conjunction with a more abundant pool of unlabeled data to enhance the model's performance. By contrast, AL approaches start with an initial set of labeled data and select the most informative data point to label

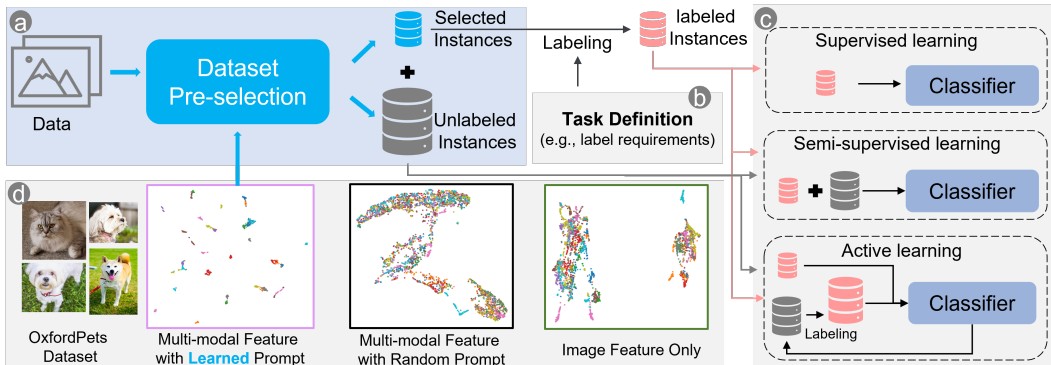

Figure 1: Our data pre-selection (a) is a novel task in data efficiency learning, and different from previous tasks, such as semi-supervised learning and active learning, which needs both seed labels and downstream task information (c). Our intuition is demonstrated by the UMAP visualization (d) of the features extracted from BLIP-2 on the OxfordPets training dataset, with each color representing the ground truth. It is evident that BLIP-2 with an appropriate prompt (purple box) can provide visually superior multimodal features compared to unimodal features (green box), which can serve as a good starting point for data pre-selection.

in an effort to maximize model performance with a limited label budget. However, existing AL methods need an initial labeled set to begin with and need to train a series of models.

Nonetheless, the task of data pre-selection poses some unique *challenges* that SSL and AL, as mentioned above, fail to adequately address. In data pre-selection, we do not have initial labeled data or know the specific downstream task. Take object detection as an example: we do not know what categories of objects need to be detected. However, both current SSL and AL methods need both initial labeled data and specific learning tasks. Therefore, this requires a method to select diverse and representative instances to cover the entire dataset with a minimum number of required labels. To achieve this, it is essential to acquire semantically meaningful, high-quality features.

Fortunately, recent advances in foundational Vision-Language (V-L) models, such as CLIP [32] and BLIP-2 [23], hold significant promise in addressing this problem. These models acquire comprehensive semantic knowledge by learning from a vast collection of image-text pairs. Consequently, they possess the ability to extract meaningful features from diverse input modalities. While there exists a numerous of studies that adapt V-L models for various tasks like few-shot classification [48], semi-supervised learning [46], and selective labeling [40], they often place emphasis on unimodal features, primarily vision, and inadvertently overlook the potential contributions of multimodal features encompassing both vision and language.

In this paper, we argue that by carefully designing the prompt, the multimodal features extracted by V-L models can offer a more powerful representation for data pre-selection. Figure 1 illustrates the unimodal and multimodal feature representations obtained from BLIP-2, with different colors representing various ground truth labels. In particular, when the class information of downstream tasks is unknown, the unimodal features become mixed, making it challenging to identify specific classes. In contrast, when coupled with an appropriate prompt, multimodal features exhibit a more scattered and distinct distribution, allowing for improved discrimination among classes. This enhanced discrimination is crucial in helping users identify the most representative and diverse instances during the data pre-selection process.

However, the process of designing an appropriate prompt can be demanding and time-consuming, often requiring trial and errors [48]. Moreover, current automatic prompt learning approaches [48, 47, 19] typically rely on a small set of labeled or pseudo labeled data, making them impractical for data pre-selection when the downstream tasks are undefined. To address this challenge, we propose a novel approach: pure unsupervised prompt learning, which aims to enhance the multimodal features extracted from V-L models for data pre-selection. Our contributions include:

- We introduce **UP-DP**, an innovative **Unsupervised Prompt** learning for enhanced **Data Pre-selection** in Vision-Language models. To the best of our knowledge, UP-DP is the

pioneering work that incorporates unsupervised prompt learning in a vision-language model specifically for data pre-selection.

- We demonstrate the robust generality of the learned prompt across different datasets, which significantly enhances feature extraction when applied in a plug-and-play manner.

- We extensively benchmark our method using various architectural settings, achieving superior performance compared to other competitors on 7 image classification datasets.

## 2 Related Works

### 2.1 Data Efficiency Learning

**Semi-supervised Learning** (SSL) combines labeled and unlabeled data to enhance learning, and can be seen as unsupervised learning with added labeled data. SSL approaches are divided mainly by how they leverage large amounts of unlabeled data. Consistency Regularization methods [33, 42] maintain model output stability under realistic perturbations like data augmentation, with UDA [42] being an example that extends supervised data augmentation to boost SSL performance. Pseudo-labeling methods [4, 41, 5] use high confidence model predictions to generate pseudo-labels for unlabeled data, training them jointly with labeled data. For example, SoftMatch [5] addresses the quantity-quality trade-off in pseudo-labeling using a truncated Gaussian function to weight samples by confidence. Transfer learning-based methods such as SimCLRv2 [6] involve supervised fine-tuning after unsupervised pre-training on unlabeled datasets, demonstrating the effectiveness of large models for SSL.

However, the success of these methods is based on the assumption of stratified sampling [4, 35], which requires equal sampling of each known class, a constraint that is often unattainable in practice. More importantly, our data pre-selection task even does NOT have any assumption for downstream tasks and also does NOT know any information about class categories.

**Active Learning** (AL) actively selects high-value instances for labeling during the iterative training process to enhance predictive performance, offering a realistic alternative to the stratified sampling setting of SSL. AL strategies generally fall into two categories: (1) methods [15, 44] that select informative instances using a scoring function such as uncertainty and (2) methods [34, 43] that choose diverse instances representing the dataset's domain. Some research [30, 1] addresses both aspects, such as ALFA-Mix [30], which evaluates label variability for perturbed instances and ensures diversity by clustering and selecting centroids.

However, current AL methods have limitations. They often require an initial labeled set, which is inefficient in low-label settings, and involve multiple rounds of labeling and training with a human annotator, making the process time-consuming. On the other hand, the requested labels are tightly associated with the training objective, necessitating distinct instances for each task. In contrast, our data pre-selection task aims for single-pass labeling applicable to any future, unknown downstream task, minimizing human effort and overall cost.

**Unsupervised Selective Labeling** (USL) [40] is the most relevant approach to ours. They proposed an SSL pipeline that avoids the unrealistic setting of stratified sampling. Specifically, they first apply unsupervised feature learning (e.g., MoCov2 [7]) to map data into a discriminative feature space. Then, they select instances for labeling for maximum representativeness and diversity in a single pass. Finally, with a strong initialization from the first step, they apply SSL to the labeled data and the remaining unlabeled data. Although their method achieves great performance and even beats stratified sampling in some settings, it is not suitable for data pre-selection. Using the same model and weights for both the selection step and the downstream task step is not a valid assumption.

### 2.2 Vision-Language Foundation Models

Vision-language models, pre-trained on large-scale image-text pairs, excel in visual representation learning. A taxonomy of V-L models [20] can be based on two aspects: (1) the expressiveness of both modalities in terms of parameters and computation, and (2) the extent of interaction within a Deep Neural Network (DNN). Early Visual Semantic Embedding (VSE) models, such as VSE++ [13] and SCAN [37], use separate feature extractors for image and text modalities, with the image extractor being more complex. They represent similarity using dot products or shallow attention layers.

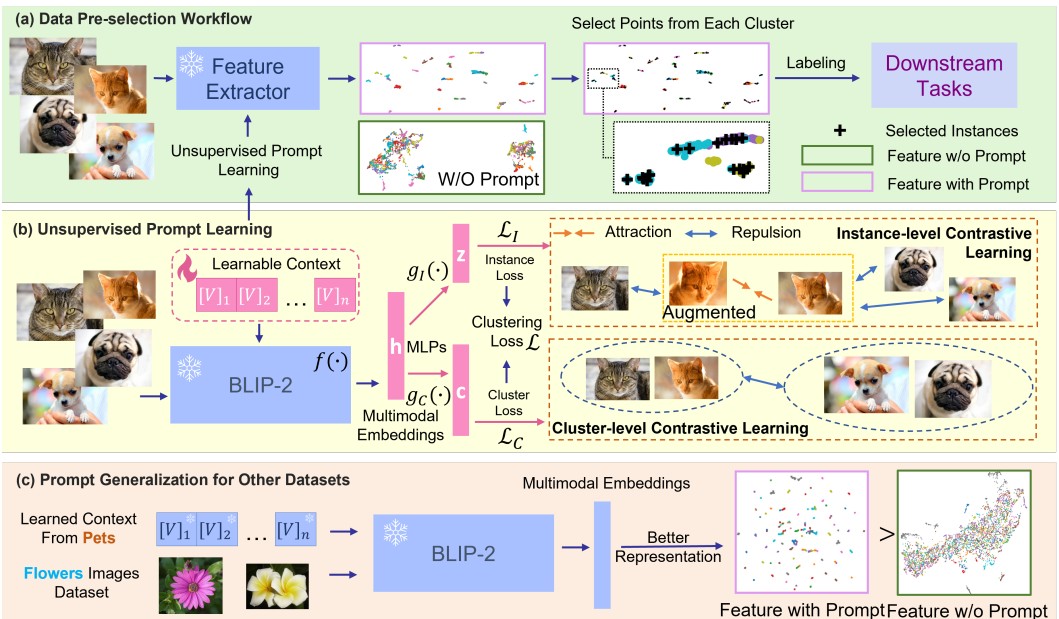

Figure 2: An Overview of Our Approach. (a) workflow for data pre-selection, (b) training process for unsupervised prompt learning, and (c) generalization of learned prompts across various datasets.

In contrast, CLIP [32] uses comparably complex feature extractors (transformers [38]) for each modality but maintains a shallow interaction. Recent models [20, 22, 39] use deep transformers for inter-modal interactions, enhancing performance in complex vision-language tasks. However, pre-training becomes computationally expensive with large feature extractors and interaction models. BLIP-2 [23] introduces a querying transformer (interaction) that utilizes pre-trained image and language models and captures robust interaction between the image and text. The interaction of two modalities allows us to effectively manage the multimodal features extracted from the V-L model merely by modifying the input text (prompt). However, determining the optimal prompt for data pre-selection remains an open question. In this work, we designed a novel approach of generating text prompts to extract multimodal features without any additional information.

## 2.3 Adaptation of Vision-Language Foundation Models

Recent advances in multimodal pre-training models have shown great success in classic vision tasks [32], prompting interest in more efficient adaptation models.

To avoid the time-consuming process of fine-tuning an over-parameterized V-L model, two main strategies have recently gained popularity: (1) V-L Adapters [16, 45], and (2) Prompt learning [48, 19]. The first category of adapters methods, like CLIP-Adapter [16] and Tip-Adapter [45], introduce lightweight networks to learn refined features for classification tasks. On the other hand, CoOp [48] proposes a continuous prompt optimization strategy for image recognition, inspired by prefix-tuning for language models [17]. However, these methods, including Unsupervised Prompt Learning (UPL) [19], mainly target classification tasks and require labels or pseudo labels, making them ill-suited for data pre-selection tasks.

Our work differs from existing literature as we approach downstream tasks with zero prior knowledge. We demonstrate the adaptation of the V-L model to extract superior features in an unsupervised manner.

## 3 Methodology

An overview of our approach is shown in Figure 2. First, we formally define the data pre-selection task. Next, we provide brief reviews on BLIP-2 and present our unsupervised prompt learning

framework, which assists BLIP-2 in extracting improved multimodal features for data pre-selection. Finally, we describe how to use the adapted BLIP-2 to select instances for downstream tasks.

## 3.1 Data Pre-selection

Figure 2a, shows the data pre-selection workflow. Suppose that we have an enormous dataset $D$ consisting of $d$ instances and an annotation budget of $l$. Our task is to select $l$ ($l \ll d$) instances for labeling, so that undefined downstream tasks trained on such a partially labeled dataset produce the best performance. Formally, let $D = (x_i, y_i)_{i=1}^d$ denote $d$ pairs of the image $x_i$ and its class label $y_i$ ($y_i = \varnothing$ if the label is unknown). Let $D_L$ denote a size $l$ subset of $D$ with known class labels. Our goal is to select $D_L \subset D$ to acquire class labels, in order to maximize the performance of an undefined model trained on labeled data $D_L$ with or without unlabeled data $D_U = D \setminus D_L$. Our data pre-selection task is challenging, as we do not have any labels to begin with. This means we do NOT know what information will make the undefined downstream task perform the best. Our idea is to select $l$ instances that are not only representative but also diverse enough to cover the entire dataset, preventing premature loss of valuable information prior to label acquisition.

The process depicted in Figure 2a illustrates the detailed data pre-selection process by extracting high-quality and semantically significant features. These features facilitate the clustering of all unlabeled images into well-structured groups. Subsequently, we select the most representative instance from each cluster for labeling. This strategy ensures both representativeness and diversity, accomplished by selecting one instance from each cluster. In the upcoming section, we will describe how we adapt BLIP-2 to this objective.

## 3.2 Unsupervised Prompt Learning

Figure 2b depicts the Unsupervised Prompt Learning approach to extract desirable features. It consists of three cooperatively learned elements: the learnable context of BLIP-2 and two Multilayer Perceptron (MLP) heads for both instance-level and cluster-level transformation. These components are jointly optimized using unsupervised clustering objectives [25], specifically instance-level and cluster-level contrastive learning. After training, the cluster assignments of all instances can be easily derived from the predicted soft labels produced by the cluster-level MLP head. When combined with the multimodal features obtained from the adapted BLIP-2, we can effectively identify the most representative and diverse instances from the entire dataset $D$.

**Bootstrapping Language-Image Pre-training** known as BLIP-2, presents an efficient pre-training strategy that leverages existing pre-trained image encoders and large language models, all while remaining frozen. By training a lightweight querying transformer (Q-Former), it specifically bridges the modality gap between images and text. In the representation learning phase, BLIP-2 connects Q-Former to a frozen image encoder and conducts image-text pair pre-training. With three image-text pre-training objectives (Image-Text Contrastive Learning, Image-grounded Text Generation, and Image-Text Matching), they establish a

---

**Algorithm 1:** Unsupervised Prompt Learning

**Input:** Unlabeled dataset $D$; training epochs $E$; batch size $N$; temperature parameters $\tau_I$ and $\tau_C$; cluster number $M$; learnable context $\mathbf{V}$; augmentation set $T$; BLIP-2 $f$; MLP heads $g_I$ and $g_C$; $D_L \leftarrow [\,]$.

**Output:** Labeled subset $D_L$

// Training
**for** *epoch* = 1 **to** $E$ **do**
    Sample a mini-batch $\{\mathbf{x}_i\}_{i=1}^N$ from $D$
    Sample two augmentations $T_a, T_b \sim T$
    // Compute instance and cluster representations:
    $\mathbf{h}_i^a = f(T_a(\mathbf{x}_i), \mathbf{V})$, $\mathbf{h}_i^b = f(T_b(\mathbf{x}_i), \mathbf{V})$
    $\mathbf{z}_i^a = g_I(\mathbf{h}_i^a)$, $\mathbf{z}_i^b = g_I(\mathbf{h}_i^b)$
    $\mathbf{c}_i^a = g_C(\mathbf{h}_i^a)$, $\mathbf{c}_i^b = g_C(\mathbf{h}_i^b)$
    Compute overall loss $\mathcal{L}$
    Update $\mathbf{V}$, $g_I$, and $g_C$ to minimize $\mathcal{L}$
**end**
// Inference
**for** $\mathbf{x}$ *in* $D$ **do**
    $\mathbf{h} = f(\mathbf{x})$ // Extract features
    $c = \arg\max(g_C(\mathbf{h}))$ // Compute cluster assignment
**end**
// Labeling
**for** $k = 1$ *to* $M$ **do**
    Select the most representative instance $\mathbf{m}_k$
    $D_L = D_L \cup \mathbf{m}_k$
**end**

---

strong connection between image and input text, i.e. forcing the model to extract visual information from the image that is most relevant to the text and automatically generating output multimodal features. This characteristic of BLIP-2 provides us with a new opportunity. By altering different prompts (input text), we can extract distinct multimodal features and find the best to enhance the process of data pre-selection.

**Learnable Prompt**   However, designing an appropriate prompt can be demanding and time-consuming, especially for data pre-selection, when we have zero knowledge of downstream tasks. Following the previous prompt learning approach, we aim to model each context token using a continuous vector that can be learned from data from end to end. Specifically, as depicted in Figure 2b, we define $N$ *learnable prompt* representation context vectors, denoted as $\mathbf{V} = \{\mathbf{v_1}, \mathbf{v_2}, ...\mathbf{v_n}\}$, each having the same dimension as the word embedding, to serve as the text input for BLIP-2. In contrast to many previous methodologies that leverage cross-entropy loss as their learning objective with label data, we have chosen to adopt unsupervised clustering as our learning objective with respect to undefined downstream tasks. Since the text encoder is differentiable, which means gradients can be back-propagated to update the context vectors. It's important to note that the base model of BLIP-2 remains unchanged during the training process.

**Instance-level Contrastive Loss**   Contrastive learning at the instance level is designed to maximize the similarities between positive instance pairs, while simultaneously minimizing those of negative pairs. Given the lack of labels for all the data, we aim to maximize the agreement between differently augmented views of the same instance and treat all other augmented instances as negative examples. Specifically, consider a randomly sampled mini-batch of image of $N$ images. Each image, denoted by $\mathbf{x}_i$, undergoes different augmentation twice, thus creating two views of the same example: $\mathbf{x}_i^a$ and $\mathbf{x}_i^b$. Subsequently, these two images are encoded, along with a learnable context $\mathbf{V}$, using BLIP-2 $f(\cdot)$, to generate multimodal representations: $\mathbf{h}_i^a$ and $\mathbf{h}_i^b$. Following this, the representations are further transformed via a nonlinear transformation MLP $g_I(\cdot)$, known as an instance-level head, which yields $\mathbf{z}_i^a$ and $\mathbf{z}_i^b$. Let $\text{sim}(.,.)$ denote the cosine similarity. Then the instance-level contrastive loss of a positive pair of examples $(i, j) \in [1, N]$ is defined as

$$\ell_i^a = -\log \frac{\exp\left(\text{sim}\left(\mathbf{z}_i^a, \mathbf{z}_i^b\right)/\tau_I\right)}{\sum_{k \in \{a,b\}} \sum_{j=1}^N \exp\left(\text{sim}\left(\mathbf{z}_i^a, \mathbf{z}_j^k\right)/\tau_I\right)}, \tag{1}$$

where $\tau_I$ is the temperature scalar. The final loss $\mathcal{L}_I = \frac{1}{2N} \sum_i^N (\ell_i^a + \ell_i^b)$, which is computed across all positive pairs in a mini-batch.

**Cluster-level Contrastive Loss**   Similar to instance-level constrative learning, cluster-level constrative learning aims to maximize the similarities between positive cluster pairs, while minimizing the negative pairs. To construct the cluster representation, we use another cluster-level head $g_C(\cdot)$ to map the multimodal representations to a $M$-dimensional space, where $M$ equals the number of clusters. Thus $m$-th element of the feature can be interpreted as the instance's probability of belonging to the $m$-th cluster. Formally, suppose with mini-batch of $N$ instances and $M$ predefined clusters, the output of the cluster head is $\mathbf{C}^a \in \mathbb{R}^{N \times M}$ under the first augmentation, thus $\mathbf{C}_{n,m}^a$ can be interpreted as the probability of instance $n$ under augmentation $a$ being assigned to the cluster $m$. The $i$-th column of $\mathbf{C}^a$ can be treated as the representation of the $i$-th cluster under augmentation $a$. Thus, we can from a positive pair of cluster by selecting $i$-th column of $\mathbf{C}^a$ and $\mathbf{C}^b$ noted as $\hat{\mathbf{c}}_i^a$ and $\hat{\mathbf{c}}_i^b$ while leaving other $2M - 2$ pair to be negative:

$$\hat{\ell}_i^a = -\log \frac{\exp\left(\text{sim}\left(\hat{\mathbf{c}}_i^a, \hat{\mathbf{c}}_i^b\right)/\tau_C\right)}{\sum_{k \in \{a,b\}} \sum_{j=1}^N \exp\left(\text{sim}\left(\hat{\mathbf{c}}_i^a, \hat{\mathbf{c}}_j^k\right)/\tau_C\right)}, \tag{2}$$

where $\tau_C$ is the cluster-level temperature parameter. To avoid a trivial solution that most instances are assigned to the same cluster, a regularization that encourages the entropy of cluster assignment probability is added: $H(\mathbf{C}) = -\sum_i^M [P(\hat{\mathbf{c}}_i^a) \log P(\hat{\mathbf{c}}_i^a) + P(\hat{\mathbf{c}}_i^b) \log P(\hat{\mathbf{c}}_i^b)]$ , where $P(\hat{\mathbf{c}}_i^k) = \frac{1}{N} \sum_{t=1}^N \hat{\mathbf{C}}_{ti}^k, k \in \{a, b\}$. The final cluster-level loss is $\mathcal{L}_C = \frac{1}{2M} \sum_{i=1}^M (\hat{\ell}_i^a + \hat{\ell}_i^b) - H(\mathbf{C})$.

**Objective Function**    The optimization is a one-stage and end-to-end process, shown in Algorithm 1. The prompt and two heads are simultaneously optimized and the overall objective function is the combination of constrastive loss at the instance and cluster level: $\mathcal{L} = \mathcal{L}_I + \mathcal{L}_C$.

**Labeling**    Finally, after training, each instance is assigned a cluster number by the adapted BLIP-2 and cluster-level MLP head. First, we directly use the probability predicted by the cluster-level head $g_C(\cdot)$ for sampling. In this case, for each cluster, the instance with the highest confidence score is selected. On the other hand we can also select medoid of each cluster to be labeled for the downstream task. We can then select the medoid of each cluster to be labeled for the downstream task. Formally given the cluster $\mathbf{X}_k$ with $N_k$ members $\{\mathbf{x}_1, \mathbf{x}_2...\mathbf{x}_{N_k}\}$, the medoid $\mathbf{m}_k$ is the instance that has the minimal average dissimilarity to all instances calculated on the multimodel feature $\mathbf{h}$ or $\mathbf{z}$, here we use $\mathbf{h}$ from $f(\cdot)$ as an example:

$$\mathbf{m}_k = \arg \min_{\mathbf{x}_i \in \mathbf{X}_k} \frac{1}{N_k} \sum_{j=1}^{N_k} (1 - \text{sim}(f(\mathbf{x}_i, \mathbf{V}), (f(\mathbf{x}_j, \mathbf{V}))). \tag{3}$$

## 4    Experiments

We assess the effectiveness of our UP-DP method in a downstream task, specifically using selected labeled instances for image classification. In particular, we perform the UP-DP on BLIP-2, strategically selecting the most representative and diverse instances for labeling, followed by the linear-probe on CLIP to compare the performance against random baselines and two variations of current state-of-the-art method: USL [40]. Lastly, we show several intriguing properties of UP-DP such as generalizability, i.e. a learned prompt from a single dataset can be directly applied to enhance the feature quality of other datasets.

**Dataset**    We select seven image classification datasets that are widely used in evaluating the V-L model adaptation approach. These datasets constitute a comprehensive benchmark, covering a diverse set of vision tasks, including the classification of generic objects (Caltech101 [14]), actions (UCF101 [36]), fine-grained categories (OxfordPets [29], FGVCAircraft [26], and Flowers102 [27]), as well as some specialized tasks such as recognizing texture (DTD [8]) and satellite imagery (EuroSAT [18]).

**Training Details**    For the base model, we use the best available vision backbone in BLIP-2, which is ViT-G. Previous work [48] on prompt learning has shown that a shorter context length can lead to better and more robust performance. Therefore, we initialize the context vectors with a fixed length of 4. The two hyperparameters, $\tau_I$ and $\tau_C$, are set to 0.5 and 1.0, respectively. Training is performed with the Adam optimizer and a learning rate of 0.0003. We optimize our model with a batch size of 256 for a total of 150 epochs on RTX 3090. We set an annotation budget of 200 for all datasets except EuroSAT, which, due to its significantly larger size (13,500 examples) and fewer classes (10), is allocated a label budget of 40 and trained for 50 epochs. The rationale for setting a uniform annotation budget of 200 images for most datasets is our lack of knowledge about the downstream task (e.g., prediction categories) in the context of data pre-selection. Thus, the annotation budget becomes our sole controllable factor. By taking into account variations in class numbers and dataset sizes, this uniform budget encompasses a a wide range of downstream task difficulties, averaging between 2 to 5 images per class.

**Baseline Methods**    As described in the related work section, USL is the most relevant approach to ours. Similarly to our method, USL also relies on semantically meaningful feature representations for each instance. We apply USL to both the image features extracted from BLIP-2 and the multimodal features extracted from BLIP-2 with the learned prompts, resulting in two settings: USL-I and USL-M. Alongside USL, we include random selection as a baseline method for the purpose of a sanity check.

**Zero-shot Recognition with BLIP-2**    In the representation learning stage, the Q-Former performs pre-training using image-text pairs with image-text contrastive learning. Similar to CLIP, after pre-training, this head can produce a similarity score between an input image and text, which can be naturally used for zero-shot recognition. We use the hand-crafted prompt provided by CoOp for evaluating zero-shot performance. As shown in the bottom row of Table 1, we observe that the

| | Method | EuroSAT | OxfordPets | DTD | Caltech101 | FGVCAircraft | UCF101 | Flowers102 | Average |
|---|---|---|---|---|---|---|---|---|---|
| RN50 | Random | 0.221 ± 0.055 | 0.193 ± 0.032 | 0.202 ± 0.016 | 0.336 ± 0.021 | 0.057 ± 0.009 | 0.113 ± 0.012 | 0.121 ± 0.015 | 0.178 |
| | USL-I | 0.309 ± 0.052 | 0.161 ± 0.020 | 0.188 ± 0.028 | 0.297 ± 0.008 | 0.048 ± 0.008 | 0.133 ± 0.013 | 0.104 ± 0.043 | 0.177 |
| | USL-M | 0.289 ± 0.044 | 0.191 ± 0.038 | 0.185 ± 0.030 | 0.286 ± 0.008 | 0.064 ± 0.010 | 0.108 ± 0.027 | 0.130 ± 0.026 | 0.179 |
| | Ours $f(\cdot)$ | 0.456 ± 0.040 | **0.229 ± 0.002** | 0.283 ± 0.011 | 0.324 ± 0.001 | 0.066 ± 0.001 | 0.184 ± 0.010 | **0.137 ± 0.003** | 0.240 |
| | Ours $g_I(\cdot)$ | 0.469 ± 0.038 | 0.214 ± 0.007 | 0.282 ± 0.007 | 0.324 ± 0.001 | 0.074 ± 0.012 | 0.179 ± 0.009 | 0.136 ± 0.003 | 0.240 |
| | Ours $g_C(\cdot)$ | **0.487 ± 0.001** | 0.204 ± 0.013 | **0.283 ± 0.007** | **0.353 ± 0.002** | **0.076 ± 0.004** | **0.205 ± 0.006** | 0.130 ± 0.001 | **0.248** |
| RN101 | Random | 0.217 ± 0.062 | 0.193 ± 0.021 | 0.139 ± 0.026 | 0.274 ± 0.023 | 0.042 ± 0.013 | 0.093 ± 0.014 | 0.116 ± 0.040 | 0.153 |
| | USL-I | 0.257 ± 0.038 | 0.149 ± 0.019 | 0.145 ± 0.049 | 0.263 ± 0.012 | 0.042 ± 0.006 | 0.101 ± 0.017 | 0.090 ± 0.042 | 0.150 |
| | USL-M | 0.209 ± 0.030 | 0.190 ± 0.052 | 0.144 ± 0.035 | 0.257 ± 0.010 | 0.058 ± 0.013 | 0.072 ± 0.037 | 0.111 ± 0.026 | 0.149 |
| | Ours $f(\cdot)$ | 0.350 ± 0.051 | **0.211 ± 0.004** | 0.250 ± 0.017 | 0.298 ± 0.001 | 0.056 ± 0.003 | 0.146 ± 0.013 | 0.133 ± 0.005 | 0.206 |
| | Ours $g_I(\cdot)$ | 0.408 ± 0.024 | 0.197 ± 0.011 | 0.245 ± 0.018 | 0.294 ± 0.006 | 0.070 ± 0.012 | 0.145 ± 0.010 | **0.133 ± 0.002** | 0.213 |
| | Ours $g_C(\cdot)$ | **0.423 ± 0.007** | 0.177 ± 0.012 | **0.260 ± 0.010** | **0.312 ± 0.002** | **0.067 ± 0.006** | **0.190 ± 0.015** | 0.120 ± 0.001 | **0.221** |
| ViTB32 | Random | 0.332 ± 0.044 | 0.340 ± 0.024 | 0.286 ± 0.049 | 0.371 ± 0.014 | 0.071 ± 0.015 | 0.150 ± 0.013 | 0.175 ± 0.024 | 0.246 |
| | USL-I | 0.422 ± 0.096 | 0.299 ± 0.038 | 0.275 ± 0.028 | 0.318 ± 0.006 | 0.074 ± 0.006 | 0.164 ± 0.020 | 0.179 ± 0.036 | 0.247 |
| | USL-M | 0.412 ± 0.077 | 0.365 ± 0.033 | 0.290 ± 0.034 | 0.310 ± 0.010 | 0.091 ± 0.008 | 0.132 ± 0.035 | 0.188 ± 0.025 | 0.255 |
| | Ours $f(\cdot)$ | 0.525 ± 0.010 | **0.439 ± 0.011** | 0.372 ± 0.002 | 0.352 ± 0.002 | 0.089 ± 0.001 | 0.214 ± 0.011 | 0.188 ± 0.008 | 0.311 |
| | Ours $g_I(\cdot)$ | 0.557 ± 0.006 | 0.429 ± 0.019 | 0.379 ± 0.002 | 0.355 ± 0.005 | 0.094 ± 0.013 | 0.207 ± 0.013 | 0.186 ± 0.006 | 0.315 |
| | Ours $g_C(\cdot)$ | **0.584 ± 0.013** | 0.380 ± 0.020 | **0.385 ± 0.007** | **0.392 ± 0.002** | **0.098 ± 0.006** | **0.234 ± 0.010** | **0.217 ± 0.003** | **0.327** |
| ViTH14 | Random | 0.482 ± 0.099 | 0.404 ± 0.042 | 0.278 ± 0.034 | 0.330 ± 0.016 | 0.118 ± 0.025 | 0.174 ± 0.024 | 0.229 ± 0.031 | 0.288 |
| | USL-I | 0.504 ± 0.103 | 0.359 ± 0.035 | 0.284 ± 0.035 | 0.298 ± 0.016 | 0.108 ± 0.010 | 0.206 ± 0.022 | 0.227 ± 0.024 | 0.284 |
| | USL-M | 0.505 ± 0.070 | 0.434 ± 0.029 | 0.304 ± 0.037 | 0.301 ± 0.015 | 0.128 ± 0.013 | 0.180 ± 0.036 | 0.221 ± 0.018 | 0.296 |
| | Ours $f(\cdot)$ | 0.577 ± 0.011 | **0.567 ± 0.005** | 0.392 ± 0.003 | 0.335 ± 0.001 | 0.116 ± 0.001 | 0.253 ± 0.017 | 0.206 ± 0.000 | 0.349 |
| | Ours $g_I(\cdot)$ | 0.596 ± 0.007 | 0.548 ± 0.025 | 0.394 ± 0.003 | 0.332 ± 0.003 | 0.131 ± 0.020 | 0.247 ± 0.005 | 0.203 ± 0.001 | 0.350 |
| | Ours $g_C(\cdot)$ | **0.634 ± 0.014** | 0.477 ± 0.021 | **0.403 ± 0.009** | **0.371 ± 0.005** | **0.143 ± 0.003** | **0.287 ± 0.010** | **0.241 ± 0.001** | **0.365** |
| ViTG14 | Random | 0.402 ± 0.045 | 0.421 ± 0.027 | 0.305 ± 0.033 | 0.334 ± 0.022 | 0.109 ± 0.014 | 0.198 ± 0.035 | 0.216 ± 0.032 | 0.284 |
| | USL-I | 0.499 ± 0.106 | 0.383 ± 0.029 | 0.285 ± 0.033 | 0.302 ± 0.013 | 0.113 ± 0.014 | 0.208 ± 0.019 | 0.243 ± 0.026 | 0.290 |
| | USL-M | 0.481 ± 0.087 | 0.461 ± 0.026 | 0.305 ± 0.038 | 0.298 ± 0.017 | 0.133 ± 0.013 | 0.184 ± 0.040 | 0.238 ± 0.024 | 0.300 |
| | Ours $f(\cdot)$ | 0.560 ± 0.024 | **0.582 ± 0.008** | 0.389 ± 0.004 | 0.339 ± 0.002 | 0.130 ± 0.005 | 0.254 ± 0.014 | 0.224 ± 0.002 | 0.354 |
| | Ours $g_I(\cdot)$ | 0.598 ± 0.004 | 0.566 ± 0.023 | 0.388 ± 0.006 | 0.336 ± 0.001 | 0.145 ± 0.021 | 0.252 ± 0.003 | 0.216 ± 0.004 | 0.357 |
| | Ours $g_C(\cdot)$ | **0.609 ± 0.019** | 0.503 ± 0.017 | **0.402 ± 0.005** | **0.376 ± 0.004** | **0.161 ± 0.005** | **0.289 ± 0.015** | **0.258 ± 0.002** | **0.371** |
| Zero Shot BLIP-2 | | 0.111 (0.100) | 0.081 (0.027) | 0.123 (0.021) | 0.379 (0.010) | 0.113 (0.010) | 0.070 (0.010) | 0.114 (0.010) | 0.141 |

Table 1: Accuracy (%) of linear probe results for CLIP trained with data pre-selected by random, USL-I, USL-M, and our method. Evaluation is carried out using five different versions of the CLIP vision encoder. The accuracy (%) of zero-shot performance with BLIP-2 is displayed in the bottom row, with the numbers in brackets representing the random guessing baseline. Our UP-DP consistently outperforms other baselines by an average accuracy around 7%.

performance of BLIP-2 is much lower than that of CLIP reported in CoOp. There are two potential reasons for this discrepancy: (1) the hand-crafted prompt for CLIP cannot be generalized for BLIP-2, and (2) the features from BLIP-2 cannot be directly generalized to these datasets, or perhaps a combination of both reasons. This highlights the necessity of adapting BLIP-2 for data pre-selection tasks.

**Linear Probe Results**  To replicate the pre-selection process, which involves separating downstream tasks from data pre-selection, we first employ UP-DP to select the most representative and diverse data points for labeling. We then apply the linear probe method using various versions of the CLIP vision encoder, ranging from ResNet-50 to ViT-G, with the labeled data. Table 1 presents the main outcomes of UP-DP in seven datasets. Overall, our UP-DP consistently outperforms other baselines with an average accuracy of 6.8% to 7.2%.

Upon further analysis of the table across different datasets, we note that UP-DP significantly outperforms the baseline on EuroSAT, yet yields mixed results on Caltech101. This variation is not entirely unforeseen considering BLIP-2's limited efficiency with complex tasks. For instance, in a zero-shot setting, it nearly resorts to random guessing (0.111 accuracy on 10 classes) on the satellite image dataset EuroSAT, whereas it exhibits a relatively superior performance (0.397 accuracy on 100 classes) on the standard object recognition dataset, Caltech101. When comparing the average performance of the second and third rows, USL-M consistently outperforms USL-I. This consistent performance gap between USL-M and USL-I further validates the effectiveness of a prompt learned by UP-DP, which assists BLIP-2 in extracting superior multimodal features.

Both USL-M and our method utilize the same multimodal features extracted from BLIP-2 with learned prompts for data selection. Essentially, USL can be seen as initially executing k-means clustering and then picking the most representative point with diversity regularization. Unlike USL-M, we explicitly train a cluster MLP head that can directly assign each instance within the cluster, where the medoid is selected in each cluster for labeling. The performance improvement between USL-M and our method showcases the effectiveness of the simple sampling strategy employed in UP-DP.

| Features | Caltech101 | DTD | FGVCAircraft | Flowers102 | OxfordPets | UCF101 | EuroSAT | Average |
|---|---|---|---|---|---|---|---|---|
| Caltech101_Prompt | 0.975 | 0.768 | 0.485 | 0.987 | 0.922 | **0.885** | 0.957 | 0.834 |
| DTD_Prompt | **0.979** | **0.809** | 0.482 | 0.990 | 0.889 | 0.878 | 0.964 | 0.836 |
| FGVCAircraft_Prompt | 0.974 | 0.768 | 0.518 | 0.986 | 0.852 | 0.862 | 0.963 | 0.829 |
| Flowers102_Prompt | 0.978 | 0.806 | **0.583** | **0.995** | **0.946** | 0.876 | 0.952 | **0.864** |
| OxfordPets_Prompt | 0.976 | 0.746 | 0.520 | 0.989 | 0.945 | 0.871 | 0.960 | 0.840 |
| UCF101_Prompt | 0.970 | 0.766 | 0.423 | 0.978 | 0.786 | 0.863 | 0.962 | 0.795 |
| EuroSAT_Prompt | 0.955 | 0.794 | 0.415 | 0.975 | 0.501 | 0.865 | **0.969** | 0.746 |
| Empty_Prompt | 0.756 | 0.501 | 0.322 | 0.719 | 0.492 | 0.744 | 0.933 | 0.606 |
| Initi_Prompt | 0.771 | 0.534 | 0.270 | 0.757 | 0.421 | 0.779 | 0.919 | 0.592 |
| Image | 0.974 | 0.763 | 0.349 | 0.984 | 0.681 | 0.875 | 0.961 | 0.760 |

Table 2: Domain Generalization Results of the Learned Prompt. The generalization capability is evaluated by the accuracy (%) of KNN classification, utilizing either image features or multimodal features extracted from BLIP-2 with various prompts. The baseline is the one with image features (in yellow). We highlight results that differ from the baseline by at least ±0.02. Green indicates higher accuracy, while red indicates lower accuracy compared to the baseline. This illustrates the impressive generalizability of the learned prompt: even when the prompts are learned from different datasets, the multimodal features almost consistently outperform the image-only features.

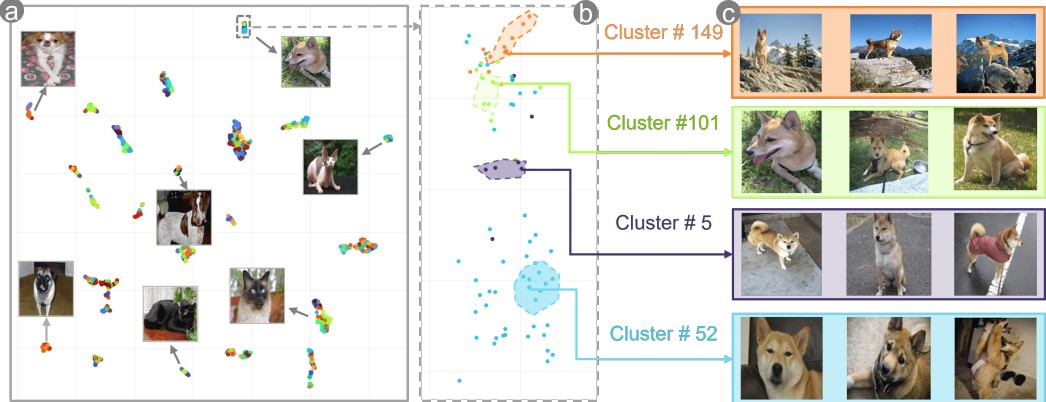

Figure 3: Visualization of Instance Features and Cluster Assignments on the OxfordPets Dataset. Color coding represents the cluster assignments determined by our cluster MLP head. The multimodal feature is extracted from BLIP-2 with learned prompts. (a) At a high level, the features extracted with the learned prompts can distinguish between various types of pets even without the labels. (b) A closer examination of one cluster reveals (c) the capability of the features to capture more detailed information, such as background and pet postures, within the same breed of Shiba Inus.

**Domain Generalization of Learned Prompt**    After establishing the effectiveness of UP-DP within a single dataset, we further demonstrate that the learned prompt has the capability to be applied across multiple datasets. This presents a considerably more challenging problem, since the fundamental features can undergo significant changes when dealing with different datasets, such as transitioning from object recognition to texture classification.

Figure 2c illustrates the way of using the prompt learned from one dataset, such as OxfordPets, to extract multimodal features from another dataset, such as Flowers102. We want to demonstrate that these features exhibit superior quality compared to features extracted directly from BLIP-2, including image features and multimodal features with empty or random prompts. We evaluate the feature quality with the testing accuracy achieved by the non-parametric K-Nearest Neighbors (KNN) classifier.

As presented in Table 2, we use the accuracy of the image feature (highlighted in yellow) as the baseline and identify results that differ from it by at least 2% (green and red). It is evident that all the results obtained from multimodal feature extraction using an undefined prompt are inferior to the image feature. Surprisingly, the multimodal feature results with the prompts learned from other datasets exhibit a significant margin of improvement over the image feature (except for the OxfordPets features extracted using the EuroSAT prompt). This demonstrates the generalizability

| EuroSAT | OxfordPets | DTD | Caltech101 | FGVCAircraft | UCF101 | Flowers102 |
|---|---|---|---|---|---|---|
| makes | shark | several | learn | diamond | healthy | learn |
| **wood** | brigham | learn | butterfly | offers | learn | **butterfly** |
| takes | saint | putting | saint | learn | attends | angel |
| healthy | laying | add | add | **plane** | performs | saint |
| ku | elegant | six | three | del | speaks | personality |
| single | chicken | three | 2020 | river | newly | picking |
| have | posing | q | 2019 | **100** | physical | adding |
| november | kate | vector | 2000 | **40** | provides | gold |
| holds | bee | che | speaks | have | choose | **spider** |
| holding | attends | some | with | get | serves | giant |

Table 3: Top Nearest Words to Learned Prompts. The selected meaningful words are obtained from the nearest words of the four context vectors learned by UP-DP. Words related to the dataset are indicated in green, while words with an orange background represent high-frequency words occurring across various datasets.

of the learned prompt. The exception for OxfordPets features can be attributed to the significant difference between the satellite image task and the fine-grained dog and cat recognition task.

**Qualitative Study**   To provide an intuitive understanding of the functioning of UP-DP, we visualize the learned feature representation and cluster assignment of all instances using 2D projection with UMAP. As depicted in Figure 3a, at high level, the features extracted from BLIP-2 with learned prompts demonstrate discriminative abilities to distinguish between different types of dogs and cats, even without any label information. Taking a closer look at the Shiba Inu group in part (b), we observe that the cluster assignments generated by our cluster-head categorize Shiba Inus into distinct clusters based on background and posture. Notably, in (c), clusters #149, #101, #5, and #52 reveal Shiba Inus against backgrounds of mountains, grasslands, concrete roads, and homes, respectively. Furthermore, Shiba Inus in the clusters #149 and #5 are typically seen standing, while those in the cluster #101 are observed lying on the ground. Our qualitative study demonstrates that UP-DP effectively selects diverse representative points from unlabeled data, resulting in improved performance for downstream tasks.

**Interpreting the Learned Prompts**   It is difficulty to interpreting the learned prompts due to their continuous vector nature. To overcome this challenge, we adopted the CoOp approach and searched the vocabulary for words that closely correspond to the learned vectors, using Euclidean distance as a metric. The search results are presented in Table 3. We observed that a few words, such as "wood" in EuroSAT (related to the forest class), certain numbers (potentially representing aircraft codes) and the word "plane" in FGVCAircraft, even "butterfly" in the Flowers102 dataset, demonstrate some relevance to their respective tasks. However, the majority of the words lack coherent meaning. This leads us to speculate that the learned vectors may encode meanings beyond the scope of the existing vocabulary. Notably, we also discovered shared words, including "learn", "saint", "add", and "attend" across different datasets, which could account for the significant generalizability of the learned prompts across various tasks.

## 5   Conclusion

In conclusion, we introduce UP-DP, a novel method that leverages Unsupervised Prompt learning to enhance Data Pre-selection with Vision-Language models. Our approach is specifically designed to achieve data efficiency without requiring prior knowledge of specific downstream tasks. By utilizing both vision and text features of V-L models, UP-DP offers a new pathway towards improving data efficiency. It outperforms existing methods by up to 20% on seven benchmark datasets and demonstrates remarkable generalizability on different datasets. We believe that our research opens promising avenues for future studies in data-efficient machine learning.

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

## Broader Impact

Our research, UP-DP, a new method for data pre-selection, holds substantial positive societal impacts. By incorporating unsupervised prompt learning into vision-language models for data pre-selection, our approach improves the efficiency of data labeling and reduces the computational resources required for machine learning tasks. By selecting a subset of instances from a large unlabeled dataset for labeling, it reduces the time and effort required for manual annotation, thereby alleviating the burden of laborious labeling tasks. This frees up human resources for more complex and creative tasks, rather than repetitive labeling work. Moreover our method democratizes access to machine learning technology. By optimizing performance with a limited annotation budget, UP-DP enables individuals or organizations with limited computational resources to train effective models. In conclusion, our work not only advances the field of data pre-selection but also has the potential to positively impact society by improving labeling efficiency, democratizing access to machine learning, and encouraging the efficient use of computational resources.

## Limitations

Our work UP-DP has demonstrated promising results in enhancing the process of data pre-selection. However, it is necessary to discuss its limitations to provide a comprehensive perspective. One limitation lies in the nature of the learned prompts. These prompts are continuous vectors, the interpretability of which is not straightforward. Unlike hand-crafted prompts that can be directly understood by humans, the meaning of continuous prompts remains opaque. This lack of transparency hinders our understanding of why and how these prompts improve the feature extraction capability of vision-language models like BLIP-2 for dataset pre-selection. Furthermore, it is even more challenging to explain the generalizability of learned prompts across different datasets, as they cover a diverse set of vision tasks. Another concern is the potential for unfairness and bias in model outcomes. UP-DP operates under a specific constraint: it selects a limited number of instances for labeling from an extensive unlabeled dataset. This strategy optimizes computational efficiency and resource allocation; however, it also introduces the risk of selection bias. If the annotation budget is extremely limited, the pre-selected data can be unrepresentative or skewed towards certain characteristics, which can inadvertently lead to biased models.

# Appendix

- Section A demonstrates the data preselection result on semantic segmentation task.
- Section B presents the ablation study on the context length.
- Section C shows the performance under large annotation budget settings.
- Section D visualizes the selected instances from different methods.

# A  Semantic Segmentation

Beyond the realm of classification tasks, we have also ventured into assessing the performance of semantic segmentation models when trained with data pre-selected via our proposed method. To elucidate, we utilized the Pascal VOC [11] dataset specifically for the semantic segmentation task, employing various adaptations of the DINOv2 model [28] to serve as our segmentation framework. The dataset comprised a total of 1454 training instances, while we constrained our annotation budget to a mere 100. As depicted in Table 4, our methodology consistently outperforms other baseline approaches in segmentation tasks, showcasing its robustness and effectiveness."

| Model | Random | USL-I | USL-M | Ours |
|---|---|---|---|---|
| Dinov2_ViTS | 54.9 | 56.4 | 58.1 | 58.5 |
| Dinov2_ViTB | 55.9 | 57.3 | 58.3 | 58.8 |
| Dinov2_ViTL | 55.2 | 54.2 | 53.7 | 57.1 |
| Dinov2_ViTG | 47.9 | 51.9 | 51.4 | 53.6 |
| Average | 53.5 | 55.0 | 55.4 | **57.0** |

Table 4: **Experiment on Semantic Segmentation Task**. The mIoU of results for Dinov2 trained with data pre-selected by random, USL-I, USL-M, and our method.

# B Context length

How many context tokens should be used? And is it better to have more context tokens? The results presented in Table 5 indicate that employing a shorter context length yields better overall performance. To investigate this hyperparameter, following the setting in CoOp, we conducted experiments on the 7 benchmark datasets using a lengthy context of 16. As illustrated in Table 5, the shorter context length consistently outperforms the longer length across all datasets and settings.

Given that our data pre-selection aims to identify instances that can be generalized to unknown downstream tasks, it becomes crucial to mitigate the overfitting caused by the longer trainable vector on the BLIP-2. This finding aligns with the analysis in CoOp, where it is argued that although having more context can potentially improve the performance of the current model, it comes at the expense of generalization.

| | Context Length | EuroSAT | OxfordPets | DTD | Caltech101 | FGVCAircraft | UCF101 | Flowers102 | Average |
|---|---|---|---|---|---|---|---|---|---|
| RN50 | 16 | $0.359 \pm 0.037$ | $0.207 \pm 0.006$ | $0.246 \pm 0.022$ | $0.312 \pm 0.002$ | $0.058 \pm 0.009$ | $0.181 \pm 0.014$ | $0.104 \pm 0.015$ | $0.210$ |
| | 4 | $\mathbf{0.456 \pm 0.040}$ | $\mathbf{0.229 \pm 0.002}$ | $\mathbf{0.283 \pm 0.011}$ | $\mathbf{0.324 \pm 0.001}$ | $\mathbf{0.066 \pm 0.001}$ | $\mathbf{0.184 \pm 0.010}$ | $\mathbf{0.137 \pm 0.003}$ | $\mathbf{0.240}$ |
| RN101 | 16 | $0.335 \pm 0.035$ | $0.188 \pm 0.017$ | $0.226 \pm 0.025$ | $0.286 \pm 0.003$ | $0.049 \pm 0.015$ | $0.132 \pm 0.012$ | $0.087 \pm 0.015$ | $0.186$ |
| | 4 | $\mathbf{0.350 \pm 0.051}$ | $\mathbf{0.211 \pm 0.004}$ | $\mathbf{0.250 \pm 0.017}$ | $\mathbf{0.298 \pm 0.001}$ | $\mathbf{0.056 \pm 0.003}$ | $\mathbf{0.146 \pm 0.013}$ | $\mathbf{0.133 \pm 0.005}$ | $\mathbf{0.206}$ |
| ViTB32 | 16 | $0.462 \pm 0.006$ | $0.400 \pm 0.023$ | $0.320 \pm 0.014$ | $0.336 \pm 0.007$ | $0.077 \pm 0.010$ | $0.212 \pm 0.019$ | $0.185 \pm 0.013$ | $0.285$ |
| | 4 | $\mathbf{0.525 \pm 0.010}$ | $\mathbf{0.439 \pm 0.011}$ | $\mathbf{0.372 \pm 0.002}$ | $\mathbf{0.352 \pm 0.002}$ | $\mathbf{0.089 \pm 0.001}$ | $\mathbf{0.214 \pm 0.011}$ | $\mathbf{0.188 \pm 0.008}$ | $\mathbf{0.311}$ |
| ViTH14 | 16 | $0.494 \pm 0.007$ | $0.478 \pm 0.018$ | $0.348 \pm 0.011$ | $0.318 \pm 0.004$ | $0.101 \pm 0.010$ | $0.249 \pm 0.018$ | $0.200 \pm 0.011$ | $0.313$ |
| | 4 | $\mathbf{0.577 \pm 0.011}$ | $\mathbf{0.567 \pm 0.005}$ | $\mathbf{0.392 \pm 0.003}$ | $\mathbf{0.335 \pm 0.001}$ | $\mathbf{0.116 \pm 0.001}$ | $\mathbf{0.253 \pm 0.017}$ | $\mathbf{0.206 \pm 0.000}$ | $\mathbf{0.349}$ |
| ViTG14 | 16 | $0.501 \pm 0.006$ | $0.507 \pm 0.015$ | $0.352 \pm 0.010$ | $0.324 \pm 0.007$ | $0.108 \pm 0.009$ | $0.251 \pm 0.019$ | $0.210 \pm 0.010$ | $0.322$ |
| | 4 | $\mathbf{0.560 \pm 0.024}$ | $\mathbf{0.582 \pm 0.008}$ | $\mathbf{0.389 \pm 0.004}$ | $\mathbf{0.339 \pm 0.002}$ | $\mathbf{0.130 \pm 0.005}$ | $\mathbf{0.254 \pm 0.014}$ | $\mathbf{0.224 \pm 0.002}$ | $\mathbf{0.354}$ |

Table 5: **Impact of Context Length.** Accuracy (%) of linear probe results for CLIP trained with data pre-selected by our method with different numbers of learnable context.

We also conducted an experiment without updating prompts, resulting in a context length of zero. The learning probe results are presented in Table 6. These findings clearly demonstrate the necessity of training the prompts. The motivation behind training the prompts/contexts is to enable the model to extract features more suited to specific datasets. If we only train $g_I$ and $g_C$ without adjusting the prompts/contexts, the image features would remain static, potentially slowing convergence and lowering performance.

| | RN50 | RN101 | ViTB32 | ViTH14 | ViTG14 | Average |
|---|---|---|---|---|---|---|
| Random | $0.119 \pm 0.021$ | $0.086 \pm 0.018$ | $0.150 \pm 0.035$ | $0.153 \pm 0.027$ | $0.178 \pm 0.028$ | $0.137$ |
| BLIP-2 w/o Prompt | $0.136 \pm 0.005$ | $0.113 \pm 0.010$ | $0.166 \pm 0.008$ | $0.176 \pm 0.003$ | $0.177 \pm 0.002$ | $0.153$ |
| BLIP-2 | $0.183 \pm 0.011$ | $0.171 \pm 0.008$ | $0.239 \pm 0.007$ | $0.259 \pm 0.004$ | $0.256 \pm 0.006$ | $\mathbf{0.221}$ |

Table 6: **Result of Training Without Prompts.** Accuracy (%) of linear probe results for CLIP, trained on the DTD dataset with data pre-selection conducted both with and without prompts. The annotation budget was set at 100.

## C  Annotation Budget

In this section, we examine the performance of data pre-selection under various annotation budgets. We conduct a linear probe experiment on CLIP, repeating it with 50% more of the available budget. Specifically, we use 60 labeled instances for EuroSAT and 300 labeled instances for all other datasets. The results are presented in Table 7. The methods (Ours and USL-M) which utilize the multimodal features extracted with learned prompt, continue to outperform other methods, showcasing the effectiveness of our unsupervised prompt learning. It is worth noting that the gain in performance is not as substantial as in scenarios with much fewer budgets before. This can be attributed to the fact that, when there is a sufficient label budget, such as 10% of the entire training dataset (Table 8), random selection can serve as a strong baseline. In contrast, our method becomes more valuable in low-budget settings.

| | Method | EuroSAT | OxfordPets | DTD | Caltech101 | FGVCAircraft | UCF101 | Flowers102 | Average |
|---|---|---|---|---|---|---|---|---|---|
| RN50 | Random | 0.323 ± 0.069 | 0.292 ± 0.023 | 0.270 ± 0.008 | 0.362 ± 0.010 | 0.062 ± 0.005 | 0.175 ± 0.041 | 0.151 ± 0.011 | 0.234 |
| | USL-I | 0.389 ± 0.042 | 0.234 ± 0.041 | 0.257 ± 0.017 | 0.349 ± 0.013 | 0.070 ± 0.007 | 0.188 ± 0.014 | 0.153 ± 0.026 | 0.234 |
| | USL-M | 0.386 ± 0.047 | 0.298 ± 0.032 | 0.269 ± 0.043 | 0.340 ± 0.021 | 0.075 ± 0.012 | 0.188 ± 0.009 | 0.133 ± 0.031 | 0.241 |
| | Ours $f(\cdot)$ | **0.398 ± 0.010** | **0.307 ± 0.007** | **0.342 ± 0.010** | **0.399 ± 0.005** | **0.084 ± 0.001** | **0.245 ± 0.006** | **0.187 ± 0.007** | **0.280** |
| RN101 | Random | 0.291 ± 0.049 | 0.251 ± 0.028 | 0.216 ± 0.031 | 0.319 ± 0.016 | 0.058 ± 0.012 | 0.142 ± 0.028 | 0.136 ± 0.018 | 0.202 |
| | USL-I | 0.325 ± 0.047 | 0.225 ± 0.047 | 0.205 ± 0.027 | 0.291 ± 0.012 | 0.066 ± 0.011 | 0.144 ± 0.014 | 0.124 ± 0.028 | 0.197 |
| | USL-M | 0.319 ± 0.062 | **0.298 ± 0.047** | 0.213 ± 0.031 | 0.294 ± 0.018 | 0.064 ± 0.018 | 0.149 ± 0.017 | 0.109 ± 0.026 | 0.207 |
| | Ours $f(\cdot)$ | **0.325 ± 0.000** | 0.294 ± 0.005 | **0.305 ± 0.010** | **0.345 ± 0.004** | **0.077 ± 0.000** | **0.224 ± 0.005** | **0.159 ± 0.005** | **0.247** |
| ViTB32 | Random | 0.438 ± 0.066 | 0.488 ± 0.070 | 0.400 ± 0.024 | 0.427 ± 0.019 | 0.085 ± 0.007 | 0.201 ± 0.011 | 0.270 ± 0.040 | 0.330 |
| | USL-I | 0.496 ± 0.039 | 0.438 ± 0.030 | 0.377 ± 0.017 | 0.393 ± 0.017 | 0.097 ± 0.010 | 0.223 ± 0.023 | 0.263 ± 0.028 | 0.327 |
| | USL-M | **0.552 ± 0.054** | **0.524 ± 0.034** | 0.409 ± 0.016 | 0.390 ± 0.021 | 0.102 ± 0.020 | 0.223 ± 0.011 | 0.237 ± 0.035 | 0.348 |
| | Ours $f(\cdot)$ | 0.541 ± 0.006 | 0.488 ± 0.019 | **0.427 ± 0.001** | **0.448 ± 0.004** | **0.104 ± 0.004** | **0.274 ± 0.003** | **0.309 ± 0.001** | **0.370** |
| ViTH14 | Random | 0.505 ± 0.110 | 0.560 ± 0.029 | 0.421 ± 0.024 | 0.425 ± 0.009 | 0.152 ± 0.012 | 0.282 ± 0.027 | 0.290 ± 0.038 | 0.376 |
| | USL-I | 0.585 ± 0.052 | 0.518 ± 0.036 | 0.408 ± 0.017 | 0.369 ± 0.015 | 0.147 ± 0.011 | 0.301 ± 0.026 | 0.313 ± 0.040 | 0.377 |
| | USL-M | **0.647 ± 0.057** | **0.629 ± 0.032** | 0.435 ± 0.020 | 0.372 ± 0.025 | **0.160 ± 0.027** | 0.292 ± 0.008 | 0.294 ± 0.039 | 0.404 |
| | Ours $f(\cdot)$ | 0.615 ± 0.028 | 0.600 ± 0.018 | **0.446 ± 0.005** | **0.435 ± 0.006** | 0.148 ± 0.009 | **0.323 ± 0.008** | **0.327 ± 0.001** | **0.413** |
| ViTG14 | Random | 0.567 ± 0.062 | 0.592 ± 0.039 | 0.440 ± 0.026 | 0.413 ± 0.008 | 0.165 ± 0.026 | 0.279 ± 0.023 | 0.341 ± 0.023 | 0.400 |
| | USL-I | 0.578 ± 0.044 | 0.546 ± 0.032 | 0.411 ± 0.017 | 0.372 ± 0.019 | 0.157 ± 0.010 | 0.312 ± 0.025 | 0.336 ± 0.037 | 0.387 |
| | USL-M | **0.627 ± 0.052** | **0.658 ± 0.031** | 0.436 ± 0.017 | 0.371 ± 0.023 | **0.171 ± 0.023** | 0.303 ± 0.004 | 0.316 ± 0.034 | 0.412 |
| | Ours $f(\cdot)$ | 0.614 ± 0.007 | 0.636 ± 0.014 | **0.445 ± 0.007** | **0.434 ± 0.005** | 0.165 ± 0.011 | **0.326 ± 0.009** | **0.342 ± 0.002** | **0.423** |

Table 7: **Impact of Annotation Budget.** Accuracy (%) of linear probe results for CLIP trained with data pre-selected by random, USL-I, USL-M, and our method after increasing 50% annotation budget.

| Dataset | Classes | Train | Val | Test | Hand-crafted prompt |
|---|---|---|---|---|---|
| EuroSAT | 10 | 13,500 | 5,400 | 8,100 | "a centered satellite photo of [CLASS]." |
| OxfordPets | 37 | 2,944 | 736 | 3,669 | "a photo of a [CLASS], a type of pet." |
| DTD | 47 | 2,820 | 1,128 | 1,692 | "[CLASS] texture." |
| Caltech101 | 100 | 4,128 | 1,649 | 2,465 | "a photo of a [CLASS]." |
| FGVCAircraft | 100 | 3,334 | 3,333 | 3,333 | "a photo of a [CLASS], a type of aircraft." |
| UCF101 | 101 | 7,639 | 1,898 | 3,783 | "a photo of a person doing [CLASS]." |
| Flowers102 | 102 | 4,093 | 1,633 | 2,463 | "a photo of a [CLASS], a type of flower." |

Table 8: **Datasets Statistics.** The detailed statistics of the 7 datasets and the hand-crafted prompts that are used for BLIP-2 zero-shot learning.

# D Visualization on Selected Instances

Figure 4(c) and (d) present the visualizations of the least-40 and top-40 instances, respectively, that were selected using our method on the EuroSAT dataset. We compare these selections with the top 40 sampled images chosen through random selection and USL. In order to facilitate understanding, we have organized the images into different rows based on their ground truth labels. Note that it is important to have a balanced selection that covers all semantic classes. However, the randomly selected instances often exhibit significant imbalances. This means that we may end up with, for example, 9 instances from Class 7 while completely missing any images from Class 5. Such imbalances are quantified using the KL divergence, which measures the distance between the current sample distribution and a uniform distribution. In this case, the KL divergence is 0.241. In contrast, our top selected instances not only provide representation from each class but also offer diversity among the selected instances. The KL divergence for our top selections is only 0.047, indicating that they are both representative and diverse. On the other hand, the 40 instances that are least likely to be selected primarily consist of outliers. These outliers have the potential to mislead downstream classifiers and introduce noise into the learning process. For instance, the first image in class 0 (AnnualCrop) may appear strikingly similar to images in Class 1 (Forest) and Class 9 (SeaLake), which can be confusing for the classification task.

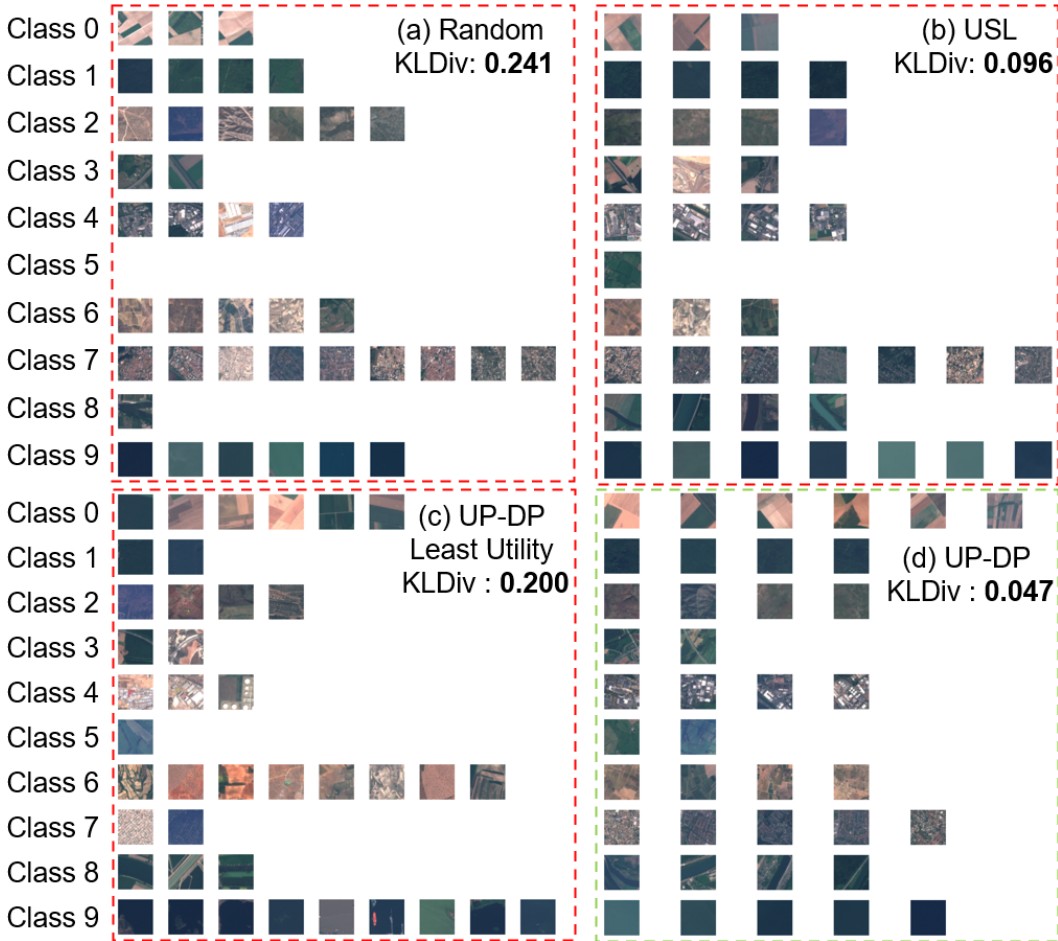

Figure 4: **Visualizations of Selected Instances from EuroSAT.** Our selection ensures balance and representativeness. In comparison, random selection can lead to imbalance and the potential omission of certain classes, particularly in low-budget settings. KLDiv represents the Kullback-Leibler divergence score between the sampling distribution and the normal distribution (stratified sampling). Classes 0-9 correspond to AnnualCrop, Forest, HerbaceousVegetation, Highway, Industrial, Pasture, PermanentCrop, Residential, River, and SeaLake, respectively.

