# OpenReview forum: "UP-DP: Unsupervised Prompt Learning for Data Pre-Selection with Vision-Language Models"
_NeurIPS.cc/2023/Conference — NeurIPS 2023 poster_

### Official Review · Reviewer_X1Uz · 2023-06-27

**Soundness:** 2 fair
**Presentation:** 3 good
**Contribution:** 2 fair
**Rating:** 5
**Confidence:** 5

**Summary:**

This paper investigates the task of data pre-selection by learning a better representation from the joint feature space of both vision and text in an unsupervised manner. The paper focuses on training text prompts to extract joint features with enhanced representation, specifically with the BLIP-2 parameters kept fixed. The aim is to achieve a diverse cluster structure that encompasses the entire dataset.

**Strengths:**

**[New task]** This paper tackles data pre-selection for labelling without accessing the information of downstream tasks, which is quite new to the community.

**[Well-illustrated figures]** The figures shown in this paper are clear enough for better understanding.

**[Good presentation]** The paper is well-written and easy to follow.


**Weaknesses:**

**[Unconvincing illustration]** In Figure 1, BLIP-2 is pre-trained with prompt and image together. This explains why using only image features yields poor performance. Consequently, the evidence presented does not convincingly demonstrate the superiority of multimodal features.

**[Need in-depth analysis]** (i) It is unclear why the self-trained model can be used for sample selection. (ii) The motivation of medoid selection is not given. It would be nice to see the rationale.

**[Missed ablation studies]** The ablation study for two hyperparameters are not given.

**[Disorganized reference format]** Please reformat the references as per some published papers.


**Questions:**

Please refer to the weaknesses part.

**Limitations:**

Please refer to the weaknesses part.

---

> ### Author Rebuttal · Authors · 2023-08-10
>
> Dear reviewer X1Uz,
>
> We would like to thank you for your valuable comments. Below is our response to your questions:
>
> Q1: “BLIP-2 is pre-trained with prompt and image together. This explains why using only image features yields poor performance. Consequently, the evidence presented does not convincingly demonstrate the superiority of multimodal features.”
>
> A1: You're correct in noting that BLIP-2 is pre-trained using both prompts and images, and I appreciate your concern regarding the demonstration of the superiority of multimodal features. The results presented in Figure 1d and Tables 1 & 2 already illustrate the superiority of using multimodal features.
>
> Quantitatively, Table 1 demonstrates that our multimodal approach ("Ours") surpasses the use of Image features (“USL-I”) in data pre-selection tasks. Table 2's KNN classification results further emphasize the superior quality of multimodal features. Specifically, utilizing Flowers102_Prompt for multimodal extraction outperforms image features by an impressive 10% in absolute value. Moreover, Figure 1d visually reinforces this advantage. It clearly displays how multimodal features offer a more distinct and scattered distribution compared to image features. This distinction significantly enhances our ability to differentiate between classes, ultimately improving the task of data preselection and KNN classification.
>
> Additionally, our method not only offers improved multimodal features but also provides an enhanced joint sampling strategy. These improvements collectively enhance the performance of the data pre-selection task, as demonstrated in Table A in the rebuttal.
>
> Q2: “It is unclear why the self-trained model can be used for sample selection. The motivation of medoid selection is not given. It would be nice to see the rationale.”
>
> A2: As demonstrated in recent literature [1][2], the first step in data selection is to obtain lower-dimensional and semantically meaningful features, which are usually acquired by models [3][4][5] using self-supervised learning. Consequently, self-trained models play a pivotal role in the data selection task.
>
> In our approach, as shown in Table 1, certain datasets (e.g. EuroSAT) prove to be effectively out-of-distribution (OOD) for BLIP-2, resulting in random guessing during zero-shot performance. To address this, we employ contrastive learning techniques to fine-tune BLIP-2's feature representation specifically for these previously unseen datasets. The enhanced feature representation subsequently leads to more accurate clustering, facilitating the identification of representative instances for the entire dataset.
>
> The concept of utilizing a medoid (or centroid) to identify a representative sample within a cluster is a widely adopted practice [6][7]. This approach is recognized for its effectiveness in selecting a sample that lies at the center of the cluster, ensuring a faithful representation. Furthermore, as highlighted in Table A, we integrate the output of the cluster head to directly sample the representative instance from each cluster, thereby adding an additional dimension to our approach. Further details can be found in the "global" response to all reviewers.
>
> Q3: “The ablation study for two hyperparameters are not given.”
>
> A3: We appreciate your observation regarding the concern about the ablation study for the two hyperparameters. We recognize this aspect and have addressed it by conducting the ablation study (Table D). Further details can be found in our response to reviewer ZwSf (A4).
>
> [1] Wang, Xudong, Long Lian, and Stella X. Yu. "Unsupervised selective labeling for more effective semi-supervised learning." European Conference on Computer Vision. Cham: Springer Nature Switzerland, 2022.
>
> [2] Xia, Xiaobo, et al. "Moderate coreset: A universal method of data selection for real-world data-efficient deep learning." The Eleventh International Conference on Learning Representations. 2022.
>
> [3] Chen, T., Kornblith, S., Norouzi, M., Hinton, G.: A simple framework for contrastive learning of visual representations. In: International conference on machine learning. pp. 1597–1607. PMLR (2020)
>
> [4] He, K., Fan, H., Wu, Y., Xie, S., Girshick, R.: Momentum contrast for unsupervised visual representation learning. In: Proceedings of the IEEE/CVF Conference on Computer Vision and Pattern Recognition. pp. 9729–9738 (2020)
>
> [5] Oord, Aaron van den, Yazhe Li, and Oriol Vinyals. "Representation learning with contrastive predictive coding." arXiv preprint arXiv:1807.03748 (2018).
>
> [6] Max Welling. Herding dynamical weights to learn. In ICML, pp. 1121–1128, 2009.
>
> [7] Sorscher, Ben, et al. "Beyond neural scaling laws: beating power law scaling via data pruning." Advances in Neural Information Processing Systems 35 (2022): 19523-19536.

---

> > ### Comment · Reviewer_X1Uz · 2023-08-15
> >
> > Thanks for the authors' response. It has solved most of my concerns so I decide to raise my score.

---

### Official Review · Reviewer_uJMt · 2023-06-30

**Soundness:** 3 good
**Presentation:** 3 good
**Contribution:** 3 good
**Rating:** 6
**Confidence:** 5

**Summary:**

The paper addresses data pre-selection (akin to active learning) problem using the highly successful vision-language models (VLMs) of late. In relation to existing approaches, the proposed approach has a few advantages, e.g., no need to have a small initial set of labeled data, no need to have multiple rounds of selection, labeling and retraining, once selected the data can be used for multiple future, unknown downstream tasks etc. The authors start with a BLIP-2 model and an unlabeled set of data $D$. The BLIP-2 model is extended to have learnable context/prompt and a few MLPs, namely instance-level and cluster-level heads. The instance-level head is employed to produce a contrastive training between two views of each unlabeled instance. Cluster-level head, on the other hand, first assigns cluster memberships to the instances and then helps to train the model with a cluster-level contrastive loss. After training for a few epochs using these two losses, cluster-level MLP along with the learned contexts/prompts are used to get the cluster assignment of unlabeled data (which can come from different downstream dataset). Finally, the medoids from each cluster form the representative selections for active learning. Experiments performed on linear probing and domain generalization show the efficacy of the proposed approach over the state-of-the-arts on benchmark datasets.

**Strengths:**

1. The use of VLMs for unsupervised active learning is appreciable. VLMs. Now-a-days, are known for good zero-shot transfer. The already well-learned representations can and did help the active learning cause.
2. The use of learnable contexts/prompts has been shown to be useful for few-shot transfer. The use of these for active learning is interesting.
3. The use of cluster-level contrastive loss cleverly avoids the use of any initial labeled set that is required in traditional cluster level losses in getting the initial clustering (akin to group contrastive loss in semisupervised literature e.g., [a]).
4. Experimental analysis and ablations show the efficacy of the proposed approach compared to sota approaches and the importance of different components of the approach as well.

[a] Singh et al., Semi-supervised action recognition with temporal contrastive learning, CVPR 2021.

**Weaknesses:**

1. One important ablation that could be useful is running the approach without the learnable prompts/contexts. What I mean is updating only $g_I$ and $g_C$ but not employing $V$ in Algorithm 1. This will help gauge the importance of the contexts/prompts vis-à-vis the instance and cluster level MLPs. Does ‘Initi_Prompt’ row in Table 2 do this?
2. Line 260+: This is more of a clarification query. When the datasets are described, I don’t see any mention of which dataset is used for pre selection. I am assuming these 7 datasets are downstream task datasets. The question is coming from Table 1. While in Table 2, it seems that the first column tells what is the dataset on which the prompts are learnt, in table 1, it is not clear. Is it that the learning is done on the same datasets on which the linear probing performances are shown for Table 1?
3. Line274+: I am not getting what is meant by 'with the learned prompts' in the baseline using USL. Does it mean everything else here is same as BLIP-2, but in addition a few prompts are learned also?

**Questions:**

Questions are already asked above (in the ‘Weaknesses’ section). Those are mostly queries for further clarification. Here in addition, let me list a few presentation related issues (typos mainly).
 - Line 32, 78, Figure 1 caption: Will these be ‘data efficient learning’ instead of ‘data efficiency learning’?
 - Line 231: ‘map’ -> ‘maps’
 - Line 244: ‘optimizing’ -> ‘optimization’
 - Line 246: ‘combine’ -> ‘combination’
 - In Figure 2a, which feature extractor is used? Is it anything different from BLIP-2?

**Limitations:**

The limitations are described well in the paper. At the same time, the authors tried to address the limitation in the supplementary material.

---

> ### Author Rebuttal · Authors · 2023-08-10
>
> Dear reviewer uJMt,
>
> We would like to thank you for your valuable comments. Below is our response to your questions:
>
> Q1: “One important ablation that could be useful is running the approach without the learnable prompts/contexts. What I mean is updating only and but not employing in Algorithm 1. This will help gauge the importance of the contexts/prompts vis-à-vis the instance and cluster level MLPs. Does ‘Initi_Prompt’ row in Table 2 do this?.”
>
> A1: Thank you for the insightful suggestion to run an ablation study without the learnable prompts/contexts. We have conducted the suggested study, and the results are presented in Table F. These findings clearly demonstrate the necessity of training the prompts. The motivation behind training the prompts/contexts is to enable the model to extract features more suited to specific datasets. If we only train g_I and g_C without adjusting the prompts/contexts, the image features would remain static, potentially slowing convergence and lowering performance.
>
> Q2: “Line 260+: This is more of a clarification query. When the datasets are described, I don’t see any mention of which dataset is used for pre selection. I am assuming these 7 datasets are downstream task datasets. The question is coming from Table 1. While in Table 2, it seems that the first column tells what is the dataset on which the prompts are learnt, in table 1, it is not clear. Is it that the learning is done on the same datasets on which the linear probing performances are shown for Table 1?”
>
> A2: Thank you for bringing up this clarification query. Your assumption is correct. In Table 1, the same dataset is used both for pre-selection and evaluation in the linear probing experiments. Let me explain the process using the DTD dataset as an example: We first train a prompt for BLIP-2 using images from the DTD dataset. This trained prompt is then used to extract features and conduct data pre-selection within the same DTD dataset. Finally, we perform a linear probe to evaluate the quality of the selected instances within the DTD dataset itself. Table 2 serves a different purpose, as it evaluates the generalization of our method by using prompts learned from one dataset to make pre-selections in another. For instance, the intersection of Caltech101_Prompt and DTD means that we use the prompt learned from the Caltech101 dataset to extract features from the DTD dataset and select instances within DTD.
>
> Q3: “Line274+: I am not getting what is meant by 'with the learned prompts' in the baseline using USL. Does it mean everything else here is same as BLIP-2, but in addition a few prompts are learned also?”
>
> A3: To perform the USL method, it requires image features. USL-I uses the image features extracted directly from the BLIP-2 model's image part without any additional training. USL-M uses multimodal features that have been learned through our proposed methods.

---

> > ### Comment · Reviewer_uJMt · 2023-08-18
> > **Post rebuttal comments**
> >
> > Thanks, authors, for the detailed response. I had a careful read of the responses to my concerns as well as to my fellow reviewers’. My clarification queries as well as the request for the ablation were addressed very good. The responses to my fellow reviewers’ queries are also apt e.g., the new experiments on additional tasks, additional backbones or linear probing results etc. I was already positive about the work and am seeing no particular reason to change.

---

### Official Review · Reviewer_soCe · 2023-07-02

**Soundness:** 3 good
**Presentation:** 3 good
**Contribution:** 3 good
**Rating:** 5
**Confidence:** 4

**Summary:**

This paper presents an unsupervised approach for data preselection, which aims to select instances for labeling from an unlabeled dataset in a single pass. The authors leverage the text features in multimodal models, specifically BLIP2, to enhance the representation for data preselection. They argue that a well-designed joint feature space of vision and text can yield improved results. To achieve this, they train text prompts to extract joint features with enhanced representation, ensuring a diverse cluster structure that covers the entire dataset. The authors employ two loss functions, namely instance-level contrastive and cluster-level contrastive, to train the learnable text prompts. These loss functions encourage the adapted multimodal model's joint representation space to be more diverse and well-separated, suitable for clustering. Experimental results on seven different datasets, along with a comparison against three baselines, demonstrate the effectiveness of the proposed approach.

**Strengths:**

I believe the strength of the paper is as follows:

- The integration of vision and language models for data pre-selection holds promise due to the added benefits of leveraging text modality.

- This paper introduces a novel approach that promotes diverse and clustered representations, addressing limitations in the current state-of-the-art BLIP2 model through the lens of prompt learning.

- The proposed method is lightweight and more efficient in terms of training costs. The paper makes a significant contribution, evident in its clear presentation and compelling results.

**Weaknesses:**

While the paper's results are strong, there are two notable aspects that could be addressed:

1. Missing CLIP baseline: Comparing the proposed approach with a CLIP baseline would provide a fundamental point of reference. Since the authors employ contrastive loss functions and prompt learning, which can be applied to any vision and language model, including CLIP as a baseline would enhance the comparative analysis.

2. Lack of integrability: A limitation of the prompt tuning method is the lack of interpretability. While prompt tuning improves model performance, it does not provide a clear explanation of why and how the model works in the combined language model embedding space. Addressing this limitation would enhance the understanding and justification of the proposed approach.

3. Performance/training-time trade-off: The paper utilizes the best-performing model of BLIPV2, which has over 7 billion parameters. However, it does not explore the performance and training-time trade-off with different BLIPV2 models. Investigating the performance of the proposed approach on BLIPV2 models with fewer parameters would provide insights into its scalability and suitability for models of varying sizes.

**Questions:**

Following questions address the weaknesses and limitations mentioned earlier. Asking the authors these questions could help improve the paper and provide a more comprehensive understanding of the proposed approach:

1. Could you compare your proposed method when built on top of CLIP as well? Including a comparison with CLIP as a baseline would provide valuable insights into the effectiveness of your approach and its advantages over a widely used vision and language model.

2. Could you provide some visualizations, such as attention weight visualizations, to explain why the trained prompts lead to better data preselection and generalization? Visualizations would enhance the interpretability of your method and shed light on the mechanisms behind its improved performance.

3. Have you replicated your proposed method on other variants of the BLIPv2 model? It would be insightful to see how your approach's performance changes when applied to BLIPv2 models with varying levels of expressiveness, particularly when the model size decreases. This analysis would help assess the scalability and adaptability of your method to different model configurations.

**Limitations:**

There are not any limitations.

---

> ### Author Rebuttal · Authors · 2023-08-10
>
> Dear reviewer soCe,
>
> We would like to thank you for your valuable comments. Below is our response to your questions:
>
> Q1: “Could you compare your proposed method when built on top of CLIP as well? Including a comparison with CLIP as a baseline would provide valuable insights into the effectiveness of your approach and its advantages over a widely used vision and language model."
>
> A1: Thank you for the suggestion to include a comparison with CLIP as a baseline. The decision not to utilize CLIP for data pre-selection in our study stems from the specific characteristics of CLIP's interaction between vision and language components, as outlined in our response to reviewer ZwSf (A5).
>
> Q2: “Could you provide some visualizations, such as attention weight visualizations, to explain why the trained prompts lead to better data preselection and generalization? Visualizations would enhance the interpretability of your method and shed light on the mechanisms behind its improved performance.”
>
> A2: Thank you for your insightful recommendation. In fact, we have already taken steps in this direction by providing visualizations and analyses of the learned prompts, as detailed in Appendix A, Table 3. We observed that certain learned words like "wood" in the EuroSAT dataset (related to the forest class), specific numbers in FGVCAircraft (possibly representing aircraft codes), and words such as "plane" and "butterfly" in different contexts demonstrate relevance to their respective tasks. However, many of the learned words lack a coherent connection to the tasks, leading us to hypothesize that the encoded meanings may extend beyond the existing vocabulary's scope. Interestingly, we also found shared words across different datasets, including "learn," "saint," "add," and "attend." These commonalities could explain the substantial generalizability of the learned prompts across various tasks.
>
> Q3: “Have you replicated your proposed method on other variants of the BLIPv2 model? It would be insightful to see how your approach's performance changes when applied to BLIPv2 models with varying levels of expressiveness, particularly when the model size decreases. This analysis would help assess the scalability and adaptability of your method to different model configurations”
>
> A3: Thank you for the valuable suggestion. We conducted additional experiments with different versions of BLIP-2 (v1 built with ViT-L and v2 built with ViT-G). The results are presented in Tables C and D. Our method outperforms other baselines using both versions, with the larger BLIP-2 v2 (with ViT-G) demonstrating the best performance.
>
> Even though BLIP-2 boasts over 7 billion parameters, our method exclusively trains a few specific components: the parameters of prompts (4 vectors), instance-head, and cluster-head. Consequently, the count of trainable parameters remains below 0.25 million, necessitating less than 60 minutes on a single GPU (RTX 3090) for training. The variation of BLIP-2 will not alter the trainable parameters, ensuring straightforward scalability.
>
> Here are more details regarding the variants of BLIP-2. It comprises two variants for the image encoder component: ViT-G and ViT-L, as well as four variants for the large language models side: OPT_2.7B, OPT_6.7B, FlanT5_XL, and FlanT5_XXL. For our data pre-selection task, we only require the Image encoder component without the language generation part, which narrows our options down to BLIP-2 with ViT-G and ViT-L.

---

> > ### Comment · Reviewer_soCe · 2023-08-20
> > **Post rebuttal comments**
> >
> > Thanks, authors for their detailed clarifications and for providing the experiment I requested. I have also reviewed the comments made by other reviewers and the authors' responses to them. The authors' rebuttal effectively addressed my concerns, so I will not be lowering my score.

---

### Official Review · Reviewer_ZwSf · 2023-07-05

**Soundness:** 1 poor
**Presentation:** 3 good
**Contribution:** 2 fair
**Rating:** 5
**Confidence:** 2

**Summary:**

The paper studies the problem of data pre-selection, which aims to select instances for labeling from an unlabeled dataset to enhance performance for downstream tasks with a limited annotation budget. The authors suggest that combining visual and textual features in a joint space can result in a better representation for data pre-selection. They introduce UP-DP, an unsupervised prompt learning approach that adapts vision-language models(specifically BLIP-2) for data pre-selection. The proposed approach outperforms the state-of-the-art on benchmark datasets and exhibits generalizability.

**Strengths:**

1. The paper introduces a novel approach for data pre-selection that incorporates unsupervised prompt learning in the vision-language model. This approach effectively exploits the multimodal features and enhances the discrimination among classes.
2. The authors provide a clear motivation for the task of data pre-selection and highlight the unique challenges it poses compared to semi-supervised learning and active learning.
3. The paper compares with the state-of-the-art on multiple benchmark datasets, demonstrating its effectiveness and superior performance. Also, it highlights the generalizability of the learned prompts across different datasets.

**Weaknesses:**

1. The authors claim that the purpose of the data pre-selection is to optimize performance for undefined diverse downstream tasks; however, they only conduct experiments on the image classification task. This limited scope is insufficient to demonstrate the effectiveness of UP-DP for various downstream tasks such as detection or segmentation. The paper would benefit from conducting more experiments on other tasks. And on top of this, it would be interesting to investigate the generalizability of the learned prompts across different tasks.
2. In Table 1, the "Zero-Shot BLIP-2" setting is unreasonable. It lacks a justifiable rationale to use the prompt for the CLIP model to evaluate the zero-shot performance of the BLIP-2 model. If the authors intend to use this baseline, they should train the learnable prompt for BLIP-2 from scratch.
3. In Table 1, the author does not include a baseline that utilizes only image features extracted from BLIP-2. By comparing with this baseline, the authors can demonstrate the efficiency of the proposed method.
4. In Table 1, the authors compare "Random", "USL-I/M", but it is suggested to compare with more approaches in the field. Including additional comparisons would strengthen the paper's evaluation and provide a better context for understanding the performance of the proposed approach.
5. The ablation studies of the proposed method are limited. It would be beneficial to conduct more comprehensive experiments to analyze the performance of UP-DP under different settings. For instance,  the authors should provide a detailed analysis of the impact of the instance-level and cluster-level contrastive loss.
6. All the experiments are carried out using the BLIP-2 model during the data pre-selection stage. However, it remains unclear whether the proposed method is exclusively effective for this particular model. It is important to consider other visual-language models, such as CLIP, to establish the efficacy of the approach.
7. The paper contains several grammar issues. For example, on page 5, line 191, it states "presents an efficient pre-training," and on page 6, line 237, it states "Thus we can from a positive pair." These sentences require revision for improved clarity and grammatical correctness.

**Questions:**

1. Is it fair to use the prompt for CLIP model to assess the zero-shot performance of the BLIP-2 model? It's supposed to train the prompt for BLIP-2 from scratch.
2. Please demonstrate that UP-DP is valid and effective for other tasks like detection and segmentation.
3. Table 1 requires revision and additional information. It should incorporate a baseline that utilizes only image features extracted from BLIP-2. Also, it is beneficial to compare with more approaches in the field.
4. Please include more ablation studies. At least, demonstrate the impact of the instance-level and cluster-level contrastive loss.
5. During the data pre-selection stage, all the features are extracted from BLIP-2 model. However, it remains unclear whether the method is exclusively effective for this specific model. It is recommended that the authors demonstrate the effectiveness of using other vision and language models as well.

**Limitations:**

The authors need to analysis the limitations of their work.

---

> ### Author Rebuttal · Authors · 2023-08-10
>
> Dear reviewer ZwSf,
>
> We would like to thank you for your valuable comments. Below is our response to your questions:
>
> Q1: “Is it fair to use the prompt for CLIP model to assess the zero-shot performance of the BLIP-2 model? It's supposed to train the prompt for BLIP-2 from scratch.” [Question context: “It lacks a justifiable rationale to use the prompt for the CLIP model to evaluate the zero-shot performance of the BLIP-2 model. If the authors intend to use this baseline, they should train the learnable prompt for BLIP-2 from scratch.”]
>
> A1: As discussed between lines 280 to 289, our primary goal isn't to benchmark against BLIP-2's zero-shot performance. Instead, we aim to emphasize the necessity of adapting BLIP-2 for data pre-selection. This is especially relevant for certain datasets like EuroSAT, which is an Out-Of-Distribution (OOD) dataset for BLIP-2, leading it to make random classifications under the zero-shot settings.
>
> Using the prompt designed from CLIP to BLIP-2 is suitable since these prompts aren’t model-specific but dataset-specific and are crafted by human experts for general zero-shot classification. As shown in Appendix C, Table 6, For example, the prompt of OxfordPets dataset is “a photo of a [CLASS], a type of pet.”
>
> Q2: “Please demonstrate that UP-DP is valid and effective for other tasks like detection and segmentation.”
>
> A2: Thank you for your insightful suggestion. In response, we have conducted additional experiments specifically for the segmentation task utilizing the PASCAL VOC dataset. As demonstrated in Table B, our UP-DP method indeed outperform other baselines in semantic segmentation tasks. More details can be found in the "global" response to all reviewers.
>
> Q3: “Table 1 requires revision and additional information. It should incorporate a baseline that utilizes only image features extracted from BLIP-2. Also, it is beneficial to compare with more approaches in the field.”
>
> A3: Thank you for the suggestions. USL-I in Table 1 is the baseline that uses image features extracted from BLIP-2, the detail is described in lines 276-279 and 307-308, which shows that our method can outperform the method only using image features.
>
> We certainly recognize the importance of comprehensive comparisons with various approaches. However, as elaborated in the Related Works section, other data efficiency approaches like active learning-based methods don't align with the specific requirements of data selection in our context, as they usually necessitate an initial labeled set and predefined downstream models. To our best understanding, USL is the most relevant approach for comparison in this study.
>
> Q4: “Please include more ablation studies. At least, demonstrate the impact of the instance-level and cluster-level contrastive loss”
>
> A4: We greatly appreciate your request for additional ablation studies, particularly regarding the impact of both instance-level and cluster-level contrastive losses. Following your valuable suggestion, we are currently conducting more in-depth ablation studies that delve into the following key aspects:
>
> 1.	Different Sampling Strategies (Table A)
> 2.	Different Versions of BLIP-2 (Table C)
> 3.	Balance and Influence of Loss Weights (Table D)
> 4.	Impact of Varied Annotation Budgets (Table E)
> 5.	BLIP-2 Training with and without Prompt (Table F)
> 6.	Difference Length of Learned Prompt (Table 4 in Appendix)
>
> In Table D, as illustrated, we've experimented with a range of weight ratios, spanning from 1:3 to 3:1, for two distinct losses within different versions of BLIP-2. The consistency of the results highlights the significance of both loss components during training. Moreover, your suggestion proved fruitful; employing a 3:1 ratio instead of a 1:1 ratio, as showcased in Table D, has demonstrated further performance enhancement for our approach.
>
> Q5: “During the data pre-selection stage, all the features are extracted from BLIP-2 model. However, it remains unclear whether the method is exclusively effective for this specific model. It is recommended that the authors demonstrate the effectiveness of using other vision and language models as well.”
>
> A5: We appreciate the recommendation to explore the effectiveness of our method with various vision and language models. The primary reason for utilizing BLIP-2 is its strong interaction between vision and language parts at a fine-grained level (namely, the tokens of image patches, and text). Through slight training adjustments to the prompt, we were able to efficiently enhance the extracted features, thereby making BLIP-2 a suitable model for data pre-selection. However, regarding “old” vision-language (V-L) models like CLIP, the interaction is limited to a cosine similarity score between the embeddings of the whole image and the full text, and this prevents us from extracting multi-modal features with fine-grained interaction between image patches and text tokens. Although we would like to extend our method to more advanced V-L models beyond BLIP-2, it currently represents the most recent and suitable V-L model.

---

### Official Review · Reviewer_ucTW · 2023-07-25

**Soundness:** 3 good
**Presentation:** 3 good
**Contribution:** 3 good
**Rating:** 5
**Confidence:** 3

**Summary:**

This paper presents a novel method to perform data preselection for the task of image classification. Data preselection refers to the task of finding the images for annotating labels and then used for training. The paper builds around the powerful visual-language model BLIP, and proposes learnable prompts as inputs to BLIP to help perform unsupervised clustering for data preselection. The paper then presents results on seven image classification benchmarks, showing the superb performance of the proposed method.

**Strengths:**

+ The presented method is a novel application of visual language model to data preselection.
+ I find the proposed learnable prompting in conjunction with unsupervised clustering novel, and as suggested by experiment effective.
+ The presented method is effective in data preselection, as demonstrated by the comparison against baseline methods.


**Weaknesses:**

- The decision to annotate 200 images per benchmark (LINE 265 - 273) seems arbitrary. Why this number? It would be great if the number can be varied and then plot the model performance accordingly to understand the effect of annotated data set size on model performance.
- USL-M, which shares the same multimodal features from BLIP-2 as the proposed UP-DP method, isn't really outperforming the baseline USL-L on EuroSAT (Table 1). Such a result contradicts the claimed effectiveness. What is the explanation?
- No results on linear probe on BLIP-2 using Random Sampling. This is needed in order to showcase the effectiveness of the proposed method.

**Questions:**

See above.

**Limitations:**

Yes

---

> ### Author Rebuttal · Authors · 2023-08-10
>
> Dear reviewer ucTW,
>
> We would like to thank you for your valuable comments. Below is our response to your questions:
>
> Q1: “The decision to annotate 200 images per benchmark (LINE 265 - 273) seems arbitrary. Why this number? It would be great if the number can be varied and then plot the model performance accordingly to understand the effect of annotated data set size on model performance.”
>
> A1: Thanks for your suggestions. We included additional results for annotation budgets from 50 to 300. As demonstrated in Table E, our method consistently outperforms other baselines regardless of the annotation budget size, thereby underscoring the robustness and effectiveness of our approach.
>
> The reason for uniformly setting it at 200 images for most datasets is that, within the context of data pre-selection, we lack knowledge about the downstream task (e.g., prediction categories). Therefore, our sole controllable factor is the annotation budget. By considering varying class numbers and dataset sizes, this uniform budget spans a wide range of downstream task difficulties, with an average of 2 to 5 images per class.
>
> Q2: “USL-M, which shares the same multimodal features from BLIP-2 as the proposed UP-DP method, isn't really outperforming the baseline USL-L on EuroSAT (Table 1). Such a result contradicts the claimed effectiveness. What is the explanation?”
>
> A2: Thanks for noting the USL-M and USL-L performance on EuroSAT in Table 1. In some instances, USL-L slightly surpasses USL-M, but given the high standard deviation from USL, it's not statistically significant. For instance, in the case of ViTG-14, the p-value is 0.77. Furthermore, when we examine the average performance across all seven datasets, USL-M consistently outperforms USL-L in five different architectures.
>
> Q3:  “No results on linear probe on BLIP-2 using Random Sampling. This is needed in order to showcase the effectiveness of the proposed method.”
>
> A3: Thank you for the suggestions. We've incorporated additional experiments that provide the performance of the linear probe on BLIP-2 (refer to BLIP-2 ViTL and BLIP-2 ViTG in Table C-F). These experiments illustrate that our approach still surpasses other baselines. It's important to note that using a linear probe on BLIP-2 contradicts the premise of data preselection, as we cannot assume that the downstream task uses the same model as the one employed in data pre-selection. This consideration is the primary reason behind our decision to utilize a linear probe on CLIP rather than on BLIP-2 in the main paper.

---

### Author Rebuttal · Authors · 2023-08-10

We extend our gratitude to all the reviewers for their meticulous comments and constructive suggestions. We are heartened by the reviewers' keen interest in our work and their recognition of its novelty, both in terms of the task and methodology.

We wish to emphasize the significance of our approach. Our proposed method not only enhances the feature representation of the dataset but also trains a cluster head capable of jointly selecting the most representative instances while encompassing the entire dataset for data pre-selection. We substantiate this through our supplementary experiment, Table A, included in the attached one-page PDF.

Appreciating the reviewers' input, we achieved further performance gains through the requested ablation studies. By exploring various sampling strategies and fine-tuning hyperparameters for the two losses, we elevated our method's performance, as evidenced in Table A and D.

Moreover, in response to reviewers' suggestions, we have conducted an experiment on the semantic segmentation task utilizing the Pascal VOC dataset [1]. We have also added more ablation studies for the classification task. These experiment results are available in the attached one-page PDF, organized as follows:

1. UP-DP Sampling Variant with Learned Heads (Table A), as requested by Reviewer X1Uz.

2. Semantic Segmentation Task Experiment (Table B), as requested by Reviewer ZwSf.

3. Impact of Different Versions of BLIP-2 (Table C), as requested by Reviewer soCe.

4. Linear Probe Result with BLIP-2 (Tables C-F), as requested by Reviewer ucTW.

5. Impact of Weights between Instance-level and Cluster-level Loss (Table D), as requested by Reviewers ZwSf and X1Uz.

6. Impact of Annotation Budget (Table E), as requested by Reviewer ucTW.

7. Impact of Training without Prompt (Table F), as requested by Reviewer uJMt.

Here is additional information about these experiments:

Table A: Beyond employing multimodal features from 'f' to select medoids to post unsupervised prompt learning, we explored multiple variants. Table A reveals that 'g_I' utilizes the projected feature from the instance head to identify medoids, while 'g_C' employs cluster probability predicted by the cluster-level head. This approach selects instances with the highest confidence score for each cluster, resulting in a significant performance enhancement.

Table B: We employed the Pascal VOC dataset for the semantic segmentation task, employing the variants of the latest DINOv2 [2] as the segmentation models. The total number of training instances was 1454, and we set the annotation budget at 100. Table B demonstrates our method's consistent superiority over other baselines in segmentation tasks.

Tables C-F: To accommodate time constraints, we chose the relatively challenging DTD dataset (with the fewest instance) for our ablation study. The annotation budget for Tables C, D, and F was set at 100, with a sampling strategy utilizing the distribution peak from cluster output by  'g_C', as previously mentioned."

[1] Everingham, Mark, et al. "The pascal visual object classes challenge: A retrospective." International journal of computer vision 111 (2015): 98-136.

[2] Oquab, Maxime, et al. "Dinov2: Learning robust visual features without supervision." arXiv preprint arXiv:2304.07193 (2023).

---

### Decision · Program_Chairs · 2023-09-21

**Decision:**

Accept (poster)

**Comment:**

The submission studies data pre-selection (i.e., select instances for labeling from an unlabeled dataset), to optimize performance for undefined downstream tasks with a limited annotation budget. Prior work relied solely on visual features. This work shows that both vision and text can yield a better representation for data pre-selection. Reviewers raised some initial concerns related to missing ablation studies. The authors provided those results. Reviewers arrived at an unanimous borderline accept recommendation. AC sees no reason to overturn an unanimous albeit borderline accept recommendation.